# Detecting Spatiotemporal Dynamics and Driving Patterns in Forest Fragmentation with a Forest Fragmentation Comprehensive Index (FFCI): Taking an Area with Active Forest Cover Change as a Case Study

**Shiyong Zhen** [1], **Qing Zhao** [1], **Shuang Liu** [1], **Zhilong Wu** [1], **Sen Lin** [1], **Jian Li** [2,*] and **Xisheng Hu** [1,*]

1    College of Transportation and Civil Engineering, Fujian Agriculture and Forestry University, Fuzhou 350002, China; 3211341025@fafu.edu.cn (S.Z.); 3211341014@fafu.edu.cn (Q.Z.); 2221327002@fafu.edu.cn (S.L.); 3201341016@fafu.edu.cn (Z.W.); 3201341024@fafu.edu.cn (S.L.)
2    College of Forestry, Fujian Agriculture and Forestry University, Fuzhou 350002, China
*    Correspondence: jianli@fafu.edu.cn (J.L.); xshu@fafu.edu.cn (X.H.); Tel.: +86-591-83706551 (J.L.); +86-591-83769536 (X.H.)

**Abstract:** Forests play an irreplaceable role in preserving soil and water, as well as realizing carbon neutrality. However, logging and urban expansion have caused widespread forest fragmentation globally, resulting in biodiversity loss and carbon emissions. Therefore, it is a prerequisite to develop a comprehensive index for evaluating the degree of forest fragmentation to propose effective policies for forest protection and restoration. In this study, a forest fragmentation comprehensive index (FFCI) was constructed through principal component analysis (PCA) based on land-use data from 2000 to 2020 in Fujian Province, composed of five commonly used landscape metrics: patch density (PD), largest patch index (LPI), mean patch area (MPA), aggregation index (AI), and division. Then, the semivariogram function and moving windows method were employed to explore the scale effect and spatiotemporal variations of FFCI. The spatial autocorrelation analysis was used to distinguish the spatial relationship of forest fragmentation, while the driving mechanisms were explored using the geographic detector (GD). The results show that the optimal scale to reflect forest fragmentation based on the semivariogram and moving window method was 3500 m. The proposed FFCI could explain more than 85% of the information for all landscape metrics, and the effectivity of FFCI was validated by urban–rural gradient and transect analysis. We also found that, despite having the highest forest coverage in China, Fujian Province has experienced severe forest fragmentation. High and medium fragmentation accounted for over 50% of all types of fragmentation, with decreasing trends in low and very low fragmentation and increasing trends in high fragmentation over time, indicating that the degree of forest fragmentation in the study area was aggravated over time. Moreover, the spatial distribution pattern of FFCI was mainly high–high clusters and low–low clusters, showing a decreasing trend year by year. The areas with high fragmentation were mainly distributed in the urban center of coastal cities, while the internal cities in western and central regions had a relatively low degree of fragmentation. Additionally, the spatial differentiation in the variation in FFCI was mainly influenced by elevation, slope, and nighttime light intensity. The superimposed impact of two factors on the variation in FFCI was greater than the impact of individual factors. These results provide an effective approach for assessing the degree of forest fragmentation and offer scientific support for mitigating forest fragmentation.

**Keywords:** forest fragmentation; landscape metrics; principal component analysis; spatiotemporal dynamic; geographic detector

## 1. Introduction

Forests play a key role in carbon sequestration, climate change mitigation, and protection of soil and water resources [1,2]. They cover 31% of the global land surface, and they

harbor more than half of the terrestrial biodiversity worldwide [3]. However, deforestation and other human activities have led to widespread forest loss globally, resulting in the appearance of large, isolated forest patches and reduced connectivity between them. In recent years, forest fragmentation has been found to widely occur [4,5], causing alarming habitat loss, climate change, and carbon loss [6,7]. In this context, it is urgent to present an appropriate index to evaluate the degree of forest fragmentation and understand its driving forces.

Over the past few decades, the advancement of remote sensing (RS) and geographic information systems (GIS) have facilitated many studies monitoring forest fragmentation [8,9]. Using high-resolution satellite imagery, the characteristics of forest patches, such as number, size, and geometry, have been employed to characterize the forest fragmentation [10,11] and investigate the spatial pattern, causes, and consequences of forest fragmentation [2,12]. FRAGSTATS offers various landscape metrics to quantify landscape configuration and composition at different levels [13], and it has been widely used to evaluate forest fragmentation [14,15]. FRAGSTATS metrics can be categorized into three types: patch-level metrics, class-level metrics, and landscape-level metrics. As forest is commonly defined as a class in landscape ecology [16], class-level metrics have been frequently used in the studies of forest fragmentation [9,17–19]. Some of these metrics quantify the magnitude or density of forest patches (e.g., patch density (PD), largest patch index (LPI), and mean patch area (MPA)), while some metrics quantify the spatial distribution or connectivity of forest class (e.g., division and aggregation index (AI)). These metrics offer information on forest fragmentation pattern and were widely used in previous studies [20–22]. However, most previous studies used single or multiple metrics to separately evaluate the degree of forest fragmentation. These metrics have their own focus, and single or multiple metrics used separately cannot comprehensively and systematically describe the overall characteristics of a forest landscape. Additionally, some metrics are strongly correlated and provide redundant information [23,24]. Therefore, it is necessary to build a comprehensive index to evaluate forest fragmentation.

Many studies have been conducted to measure the driving force of forest fragmentation, which includes socioeconomic, natural, and anthropogenic factors. Human activities and socioeconomic change have a great impact on forest fragmentation in many regions [25,26], and most studies employed population and nighttime light intensity as proxies. In additional, natural factors such as altitude, temperature, and precipitation determine the environmental condition of forests and are also considered to be related to forest fragmentation [27]. However, forest fragmentation is a complicated evolution process, and it may be influenced by the interaction of many factors [28]. In this context, the geographic detector (GD) [29], a statistical method based on the theory of spatial stratified heterogeneity, was used to analyze the relationship between forest fragmentation variation and driving factors in this study. In addition to analyzing the explanatory power of single driving factors, GD can quantitatively detect the interactions between driving factors [30].

Located in the southeast coast of China, Fujian Province has a subtropical climate, and its terrain mainly features mountains and hills. It is one of four major forest areas in China. Similar to other provinces in China, a series of reforestation projects were implemented in Fujian, such as the "National Forestation Program", "Natural Forest Conservation Program", and "Sloping Cropland Conversion Program" [31], leading to a constant increase in forest coverage. However, numerous studies showed that forest cover increase does not denote a decrease in forest fragmentation [32–34]. Previous studies indicated that Fujian Province is one of the most active forest cover change regions in China [35,36]; frequent changes in forest cover can lead to an increase in forest fragmentation.

In this context, this study aimed to construct a comprehensive index to evaluate the extent of forest fragmentation; the aims of this study were to (1) identify spatiotemporal variations characteristics of forest fragmentation in Fujian, (2) determine the dominant driving factors influencing variations in forest fragmentation, and (3) identify the impact of the interactions between driving factors.

## 2. Materials and Methods

### 2.1. Study Area

Fujian Province (23°30′–28°19′ N, 115°50′–120°47′ E) is located on the southeast coast of China, with an area of 121,400 km² (Figure 1). The terrain is high in the northwest and low in the southeast. The province is dominated by mountains and hills (over 80%). It has an irregular coastline with numerous bays and islands. The climate is mild and humid, with an annual average temperature of 16.3–23.3 °C, and an average annual precipitation of 800–1900 mm. The province is rich in forest and plant resources; the forest coverage rate of Fujian Province is as high as 66.8% in 2022, ranking first in China for 43 consecutive years. The main vegetation type is subtropical evergreen broad-leaved forest and subtropical monsoon rain forest.

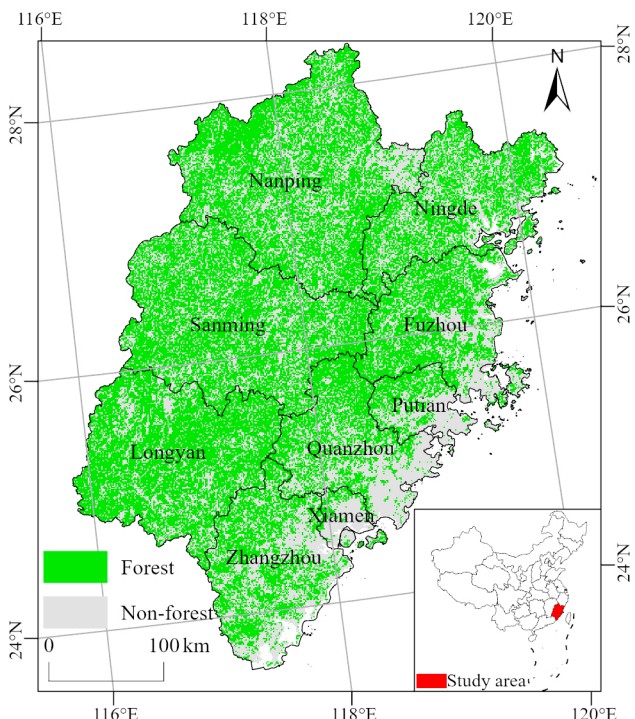

**Figure 1.** Location of study area and status of forest cover in 2020.

### 2.2. Data Sources and Processing

#### 2.2.1. Land-Use Data

The land-use/cover change (LUCC) data (including 2000, 2010, and 2020) with a spatial resolution of 30 m were obtained from Data Center for Resource and Environmental Science, Chinese Academy of Sciences (http://www.resdc.cn/) (accessed on 12 October 2021). The dataset was constructed by human-computer interaction and visual interpretation technology based on Landsat-TM/ETM+ and Landsat 8 satellite remote sensing images data, with an accuracy of more than 93% [37]. The data divide land use into six primary types (cultivated land, forest, grassland, water body, artificial land, and unused land) and 25 secondary types.

#### 2.2.2. Driving Factor Data

Table 1 lists all the driving factors selected in this study, which included geomorphic, anthropogenic, and socioeconomic factors. The digital elevation model (DEM) from the ASTER GDEM V3 product with a spatial resolution of 30 m (https://www.gscloud.cn/) (accessed on 6 December 2021) was used to extract elevation and slope data through the Surface tool in ArcGIS 10.3 software. The population density data were taken from the WorldPop dataset (https://www.worldpop.org/) (accessed on 12 December 2021)

with a spatial resolution of 1000 m. The meteorological data with a spatial resolution of 1000 m were taken from the National Earth system Science data Center (http://www.geodata.cn/) (accessed on 12 December 2021), including monthly average temperature and monthly precipitation in 2000, 2010, and 2020. The annual mean temperature and annual precipitation data were calculated using the Raster Calculator tool in ArcGIS 10.3. The nighttime light intensity data with a spatial resolution of 1000 m were taken from the National Earth System Science Data Center (http://www.geodata.cn/) (accessed on 12 December 2021), representing the socioeconomic information. All data were converted into the Albers coordinate system.

**Table 1.** List of driving factors.

| Type | Data | Variable |
|---|---|---|
| Geomorphic factors | DEM | Elevation |
|  |  | Slope |
| Meteorological factors | Monthly average temperature | Annual mean temperature variation (AMTV) |
|  | Monthly precipitation | Annual precipitation variation (APV) |
| Socioeconomic factors | Population density | Population density variation (PDV) |
|  | Nighttime light intensity | Nighttime light intensity variation (NLIV) |

*2.3. Calculation of Forest Fragmentation Comprehensive Index*

2.3.1. Selection of Landscape Metrics

Forest fragmentation refers to the process of subdividing the original large and continuous extensions of forests into relatively small and isolated patches [38]; therefore, the landscape metrics which can characterize the size, shape, and distribution of forest patches were mostly selected to characterize forest fragmentation in previous studies [39,40]. In this study, we used FRAGSTATS 4.2 software to calculate the following landscape metrics to construct the forest fragmentation comprehensive index: (1) patch density (PD), the ratio of the number of forest patches and the total area; (2) largest patch index (LPI), the proportion of the largest forest patch and the total area; (3) mean patch area (MPA), the average area of all patches of forest class; (4) aggregation index (AI), the frequency of side by side appearance of forest patches on the landscape; (5) division, the spatial proximity of forest patches.

2.3.2. Construction of Forest Fragmentation Comprehensive Index

In this study, we aimed to construct a comprehensive index (i.e., FFCI) to evaluate forest fragmentation. On the basis of the abovementioned landscape metrics, we used principal component analysis to synthesize the FFCI and compute the weight of each metric for the FFCI. Principal component analysis (PCA) is a multidimensional factor compression technique, which can be employed to reduce redundant information of landscape metrics [21]. It analyzes the variance of a set of correlated variables and transforms them into a new set of uncorrelated independent components (principal components, PCs). Additionally, the weight of each variable can be calculated automatically according to its contribution to the principal components, which provides more objective and robust results compared to artificially determining weights.

The process of PCA was accomplished using the PCA Rotation tool in ENVI. All the metrics were normalized to 0–1 before PCA. The percentage variance of PC1 was larger than 85%, indicating that PC1 represented most of the information from the original landscape metrics. Thus, PC1 was used to construct FFCI images for all study years.

*2.4. Moving Window and Semivariogram Method*

To get the optimal analysis scale of landscape metrics in the study area, we calculated three landscape metrics (division, LPI, and PD) at 10 moving window sizes (from 500 m to 5000 m with an interval at 500 m) through the moving window function of FRAGSTATS

4.2 software. The semivariogram can reveal the spatial heterogeneity of variables by measuring the relationship between the degree of variation and the distance between the spatial attributes of two points. The spherical model is a function for fitting in the semivariogram, and it contains three fitting indicators: nugget value (C), sill value $(C_0 + C)$, and range value $(A_0)$. The nugget/sill ratio $[C/(C_0 + C)]$ reflects the degree of the spatial variability caused by the random part in the total variation [41]. A higher value denotes more obvious spatial autocorrelation. When the value reaches a relatively stable level, the scale can be considered a suitable window size for the landscape metrics of the study area [42]. In this study, we used ArcGIS 10.3 software to generate 10,000 random points in the study area and extracted the three calculated landscape metrics at 10 moving window sizes into these random points. The semivariogram model in GS+ 9 software was used to measure the change in landscape metrics and determine the optimal characteristic scale. The value of the semivariogram ($\gamma$(h)) was calculated using Equation (1).

$$\gamma(h) = (1/2N(h)) \sum_{i=1}^{N(h)} [Z(x_i) - Z(x_i + h)]^2,  \tag{1}$$

where $\gamma$(h) is the semivariogram at the lag distance of h, N(h) is the number of pairs separated by lag h, and $Z(x_i)$ and $Z(x_i + h)$ are the data values at position $x_i$ and position $x_i + h$.

### 2.5. Spatial Autocorrelation Analysis

Spatial autocorrelation analysis is used to reveal correlations of attribution values between target units and neighboring target units, including global spatial autocorrelation and local spatial autocorrelation [43]. Moran's I value is the most used coefficient of spatial autocorrelation analysis. In this study, to obtain the optimal analysis scale of the FFCI, global spatial autocorrelation analysis was used to analyze the spatial autocorrelation of FFCI at different spatial scales, and the global Moran's I value of FFCI was calculated at different spatial scales using Equation (2).

$$\text{Global Moran's I} = \frac{n\sum_{i=1}^{n}\sum_{j=1}^{n} w_{ij}\left(x_i - \bar{x}\right)\left(x_j - \bar{x}\right)}{S_0\sum_{i=1}^{n}\left(x_i - \bar{x}\right)^2},  \tag{2}$$

where n is the total number of spatial units at different spatial scales in the study area, $w_{ij}$ is the spatial weight between unit i and unit j, $x_i$ and $x_j$ are the FFCI values of unit i and unit j, $\bar{x}$ is the average FFCI value in all spatial units in the study area, and $S_0$ is the sum of all spatial weights. The global Moran's I value varies from $-1$ to 1. A positive value of Moran's I represents a positive spatial correlation, while a negative value of Moran's I represents a negative spatial autocorrelation; a zero value of Moran's I represents a random spatial distribution.

Furthermore, local Moran's I analysis is used to assess the spatial aggregation characteristics (i.e., hot spots and cold spots) of variables in local areas. Local Moran's I value was calculated using Equation (3), and it was employed in the LISA (local indicator of spatial association) diagram to identify the aggregation type of FFCI in the GeoDa 1.8 software. Clusters are categorized as "high–high clusters (hot spots)", "high–low clusters, "low–high clusters", and "low–low clusters (cold spots)" [44].

$$\text{Local Moran's I} = \frac{x_i - \bar{x}}{\sum_{i=1}^{n}\left(x_i - \bar{x}\right)^2} \sum_{j=1}^{n} w_{ij}\left(x_j - \bar{x}\right).  \tag{3}$$

*2.6. Geographical Detector*

GD is a statistic method to measure the hierarchical spatial heterogeneity and driving mechanism of geographic factors [45]. It measures the linear and nonlinear statistical relationship between the dependent and the independent variables by comparing the variance between them. It includes four detector modules, factor detector, interactive detector, ecological detector, and risk detector, which are respectively used to analyze the relative importance of different factors, significant differences, interaction mechanism, and significant value distribution range, the first two of which were used in this study.

The factor detector can measure the explanatory power of driving factors, which was used to reveal the contribution of explanatory variable to the spatial differentiation of variation in FFCI in this study. It determines the explanatory power of the driving factors by comparing the mathematical statistical relationship between the total variance of the whole region and the intralayer variance. The q value is used to measure the interpretation intensity, calculated using Equation (4).

$$q = 1 - \frac{\sum_{h=1}^{L} N_h \sigma_h^2}{N \sigma^2},$$ (4)

where q is the explanatory power of the driving factor, h is the strata number of the FFCI or a certain driving factor (h = 1, 2, ..., L), $N_h$ and N are the number of samples in strata h and the whole area, $\sigma_h^2$ is the variance in the value of the FFCI in strata h, and $\sigma^2$ is the variance of the FFCI value in the entire study area.

The interactive detector is used to judge whether the explanatory power of interaction between two driving factors is enhanced or weakened. The q value of the interaction was compared to assess whether there was an interaction between two factors with regard to the variation in FFCL. The interaction types and judgment basis are shown in Table 2.

**Table 2.** Interaction types and judge basis.

| Judge Basis | Interaction Types |
|:---:|:---:|
| $q(x_1 \cap x_2) < \text{Min}(q(x_1), q(x_2))$ | Nonlinear weakening |
| $\text{Min}(q(x_1), q(x_2)) < q(x_1 \cap x_2) < \text{Max}(q(x_1), q(x_2))$ | Single-factor nonlinear weakening |
| $q(x_1 \cap x_2) > \text{Max}(q(x_1), q(x_2))$ | Double-factor enhancement |
| $q(x_1 \cap x_2) = q(x_1) + q(x_2)$ | Independence |
| $q(x_1 \cap x_2) > q(x) + q(x_2)$ | Nonlinear enhancement |

## 3. Results

### 3.1. Scale Effects of Semivariogram Analysis

Figure 2 showed the nugget/sill ratio of three landscape metrics (division, LPI, and PD) at different window sizes. When the window size was less than 3500 m, the nugget/sill ratio of each landscape metrics increased with the increase in window size, indicating that the degree of spatial variability was high and the spatial autocorrelation was not obvious. The nugget/sill ratio of each landscape metrics tended to be stabilized when the window size was 3500 m, which indicates that the spatial autocorrelation became obvious and spatial variability was low. Therefore, the scale of 3500 m was selected as the optimal moving window size to reflect the spatial variability characteristics of landscape metrics.

### 3.2. Principal Component Analysis

On the basis of the obtained results, five landscape metrics (AI, LPI, MPA, division, and PD) were calculated with a moving window size of 3500 × 3500 m in FRAGSTATS 4.2 software. The principal component results of the five landscape metrics obtained by principal component analysis are shown in Table 3. The results showed that PC1 explained more than 85% of total variance for each study year, indicating that PC1 integrated most of the information and characteristics of the five landscape metrics; therefore, PC1 was proposed to construct the FFCI to characterize the degree of forest fragmentation in the

study area, with the weights of AI, LPI, MPA, division, and PD being 0.599, 0.569, 0.175, 0.070, and 0.531 in 2000, 0.600, 0.526, 0.170, 0.071 and 0.574 in 2010, and 0.601, 0.525, 0,160, 0.577, and 0.071 in 2020, respectively. The resulting FFCI values were standardized within a range of 0 to 1, with higher values indicating high level of forest fragmentation, and lower values indicating low level of forest fragmentation.

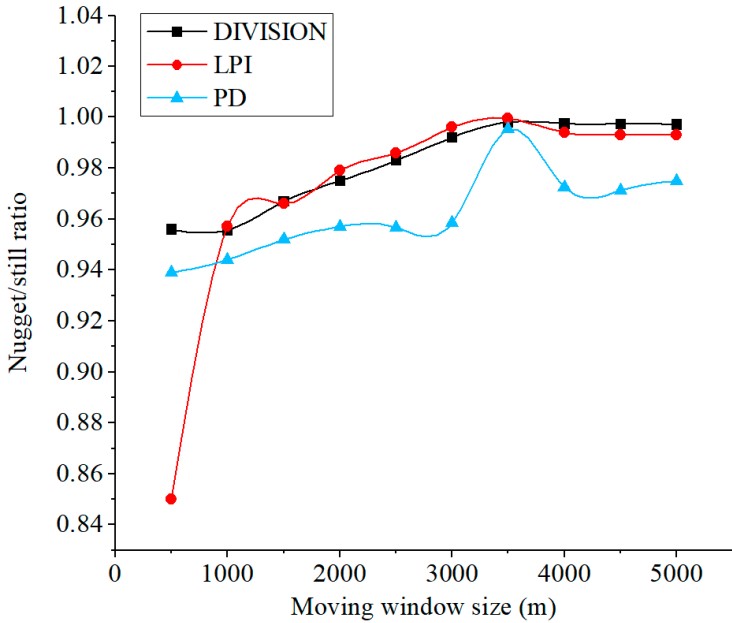

**Figure 2.** The nugget/sill ratio changes of landscape metrics with different moving window sizes.

**Table 3.** Results of the principal components analysis.

| Year | Index | PC1 | PC2 | PC3 | PC4 | PC5 |
|------|-------|-----|-----|-----|-----|-----|
| | AI | 0.599 | −0.300 | 0.128 | 0.731 | 0.014 |
| | LPI | 0.569 | −0.479 | −0.001 | −0.661 | −0.101 |
| | MPA | 0.175 | 0.125 | −0.966 | 0.079 | −0.124 |
| 2000 | Division | 0.070 | −0.031 | −0.123 | −0.068 | 0.987 |
| | PD | 0.531 | 0.815 | 0.190 | −0.134 | 0.003 |
| | Eigenvalue | 0.091 | 0.011 | 0.002 | 0.0001 | 0.0002 |
| | Percentage variance (%) | 86.7 | 10.1 | 2.3 | 0.7 | 0.2 |
| | AI | 0.600 | −0.293 | 0.122 | 0.734 | 0.026 |
| | LPI | 0.526 | 0.819 | 0.188 | −0.134 | 0.001 |
| | MPA | 0.170 | 0.125 | −0.966 | 0.076 | −0.125 |
| 2010 | Division | 0.071 | −0.032 | −0.125 | −0.085 | 0.985 |
| | PD | 0.574 | −0.477 | 0.003 | −0.656 | −0.113 |
| | Eigenvalue | 0.089 | 0.010 | 0.002 | 0.0007 | 0.0002 |
| | Percentage variance (%) | 86.7 | 10.1 | 2.2 | 0.7 | 0.2 |
| | AI | 0.601 | −0.290 | 0.124 | 0.734 | 0.024 |
| | LPI | 0.525 | 0.822 | 0.175 | −0.135 | 0.002 |
| | MPA | 0.160 | 0.118 | −0.967 | 0.084 | −0.138 |
| 2020 | Division | 0.577 | −0.475 | −0.003 | −0.656 | −0.110 |
| | PD | 0.071 | −0.031 | −0.139 | −0.080 | 0.984 |
| | Eigenvalue | 0.089 | 0.010 | 0.002 | 0.0007 | 0.0002 |
| | Percentage variance (%) | 86.9 | 10.2 | 2.0 | 0.7 | 0.2 |

### 3.3. Spatiotemporal Characteristics of FFCI

To facilitate the analysis of spatiotemporal variations of forest fragmentation, the FFCI obtained by principal component analysis was classified into five categories (Figure 3):

very low ($0 \leq$ FFCI $< 0.2$), low ($0.2 \leq$ FFCI $< 0.4$), medium ($0.4 \leq$ FFCI $< 0.6$), high ($0.6 \leq$ FFCI $< 0.8$), and very high ($0.8 \leq$ FFCI $< 1.0$). The results showed that the areas with lower fragmentation were mainly distributed in Longyan, the west of Nanping, and Ningde, the areas with medium to high fragmentation were mainly distributed in Sanming, Putian, and Zhangzhou, and the areas with higher fragmentation gathered in the central urban area of coastal cities. Table 4 listed the proportion of each type of FFCI in 2000, 2010, and 2020. The results show that high-fragmentation areas had the highest proportion in all three periods (27.7%, 28.1%, and 27.7%), followed by medium-fragmentation areas. The sum of the proportions of these two fragmentation types exceeded 50% in all 3 years. The proportion of low fragmentation was the lowest, which indicated that the degree of forest fragmentation in the study area was at an upper–middle level. From 2000 to 2020, the very-low- and low-fragmentation areas showed a gradual decrease. The proportion of medium-fragmentation areas decreased from 27.4% in 2000 to 27.1% in 2020; however, the value was moderately increased to 27.6% in 2010. The proportion of high-fragmentation areas also showed a slight increase from 27.7% in 2000 to 28.1% in 2010, but then decreased to 27.7% in 2020. The areas with very high fragmentation showed a constant increasing trend throughout the study periods, with a higher increase in the 2010–2020 period than in the previous period. Thus, the degree of forest fragmentation in study area increased over time.

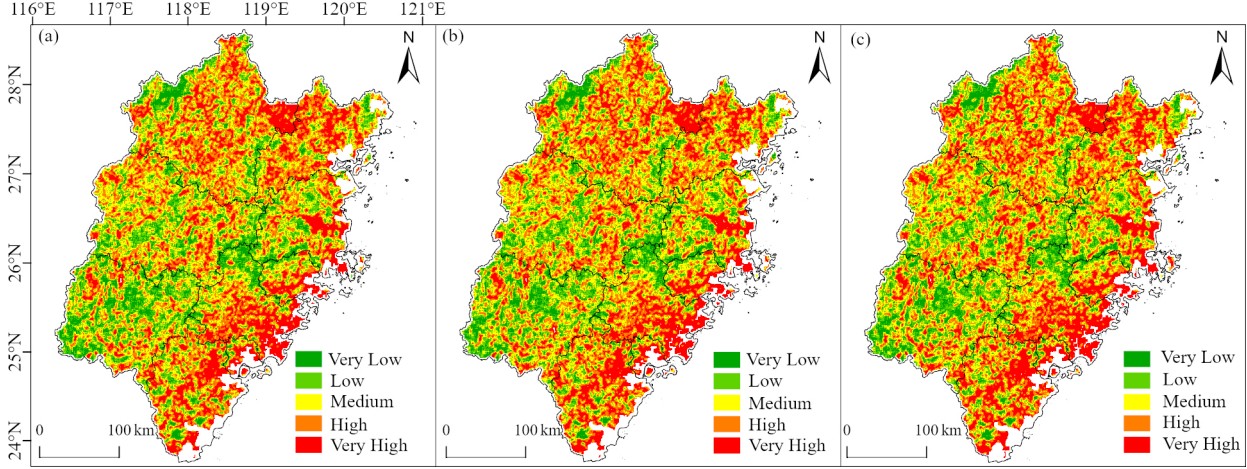

**Figure 3.** Spatial distribution of FFCI: (**a**) 2000; (**b**) 2010; (**c**) 2020.

**Table 4.** Proportions of various types of forest fragmentation from 2000 to 2020.

| Fragmentation Type | 2000 (%) | 2010 (%) | 2020 (%) |
|:---:|:---:|:---:|:---:|
| Very low | 9.1 | 8.6 | 8.3 |
| Low | 17.8 | 17.5 | 16.9 |
| Medium | 27.4 | 27.6 | 27.1 |
| High | 27.7 | 28.0 | 27.7 |
| Very high | 18.0 | 18.3 | 20.0 |

The temporal variation in the FFCI and the proportion of forest fragmentation change types in two study periods (2000–2010 and 2010–2020) are shown in Figure 4 and Table 5. In the periods from 2000 to 2010, the proportions of the three types of forest fragmentation change types from large to small were in the order unchanged > increased > decreased. The areas with decreased fragmentation accounted for 2.7%, which were mainly distributed in Ningde and Sanming. The areas with increased fragmentation accounted for 4.8%, which were mostly distributed in Longyan, Zhangzhou, Quanzhou, and Putian. The unchanged areas had the largest proportion (92.5%) in the study area, indicating that the change in degree of forest fragmentation was small in most areas. From 2010 to 2020, the proportion

of areas with increased and decreased forest fragmentation both showed an increasing trend compared to the last stage, and their spatial distribution was relatively scattered, indicating that the degree of forest fragmentation in the study area increased.

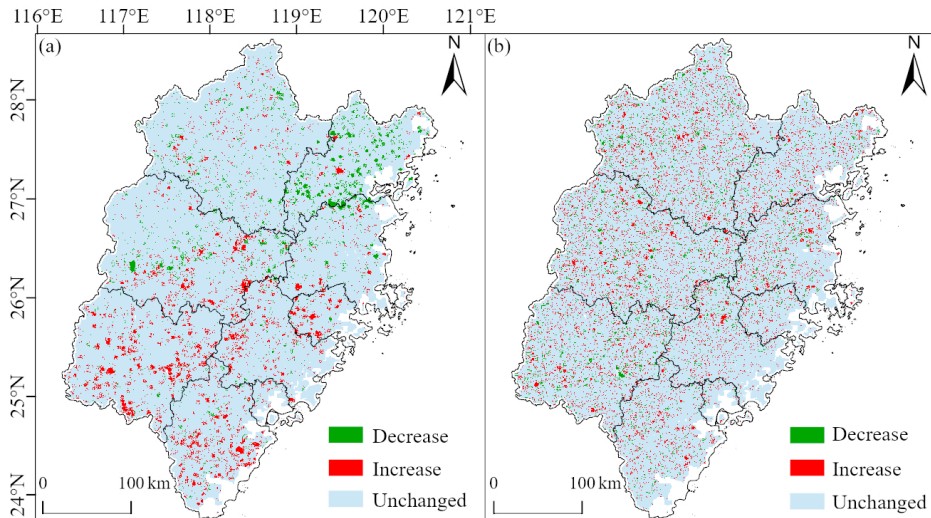

**Figure 4.** Changes in FFCI from 2000 to 2020: (**a**) 2000–2010; (**b**) 2010–2020.

**Table 5.** Proportions of forest fragmentation change types from 2000 to 2020.

| Period | Decrease (%) | Unchanged (%) | Increase (%) |
|---|---|---|---|
| 2000–2010 | 2.7 | 92.5 | 4.8 |
| 2010–2020 | 2.8 | 90.5 | 6.7 |

### 3.4. Analysis of Spatial Cluster of FFCI

Figure 5 shows that the Moran's I values of FFCI at different spatial scales were all >0 and the z-scores were all >2.58 (statistically significant at the 1% level). Among them, the Moran's I value of spatial scales from 1000 m to 5000 m was higher, which was 0.731, 0.421, 0.296, 0.305, and 0.261 respectively, and the largest values of Moran's I in the three periods were all at the scale of 1000 m; therefore, 1000 m was selected as the analysis scale of the spatial cluster pattern of FFCI.

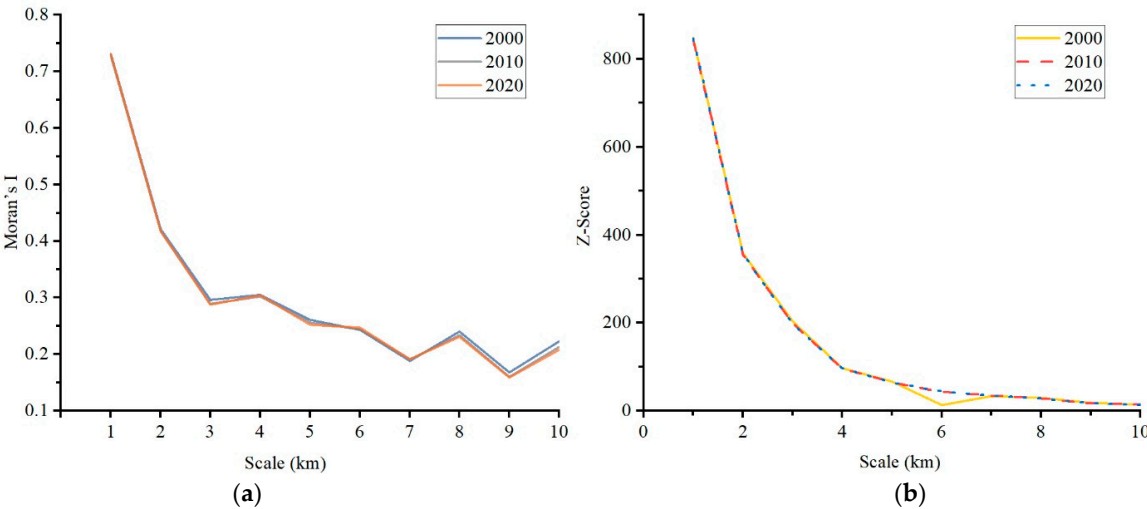

**Figure 5.** Global autocorrelation parameters of FFCI at different spatial scales: (**a**) Moran's I value; (**b**) z-Score.

The analysis of local spatial autocorrelation of FFCI is shown in Figure 6. The spatial distribution of the FFCI was dominated by high–high and low–low clusters, with limited areas featuring high–low and low–high clusters. The high–high clusters representing the agglomeration areas with a high value of the FFCI were mainly distributed in Nanping and Ningde, mostly concentrating in the central urban areas of coastal cities. Low–low clusters representing the agglomeration areas with a low value of the FFCI were primarily distributed in Longyan, Wuyishan, and the junction of Quanzhou and Fuzhou. According to Table 6, in the period from 2000 to 2010, the areas with high–high clusters in Fuzhou slightly decreased, but expanded to varying degrees in other regions, with Zhangzhou experiencing the greatest increase and Sanming experiencing the smallest. In Fuzhou, Nanping, Sanming, Xiamen, and Zhangzhou, the areas with low–low clusters showed an expansion trend, but contracted in other areas. From 2010 to 2020, the areas with high–high clusters decreased in Putian, Sanming, Xiamen, and Zhangzhou, but expanded elsewhere. Meanwhile, the low–low clusters areas expanded in Longyan, Ningde, Quanzhou, and Xiamen, with the most significant increase in Ningde and the most significant decrease in Nanping, Fuzhou, Putian, Sanming, and Zhangzhou. There was no significant variation in the areas with high–low and low–high clusters during either study period.

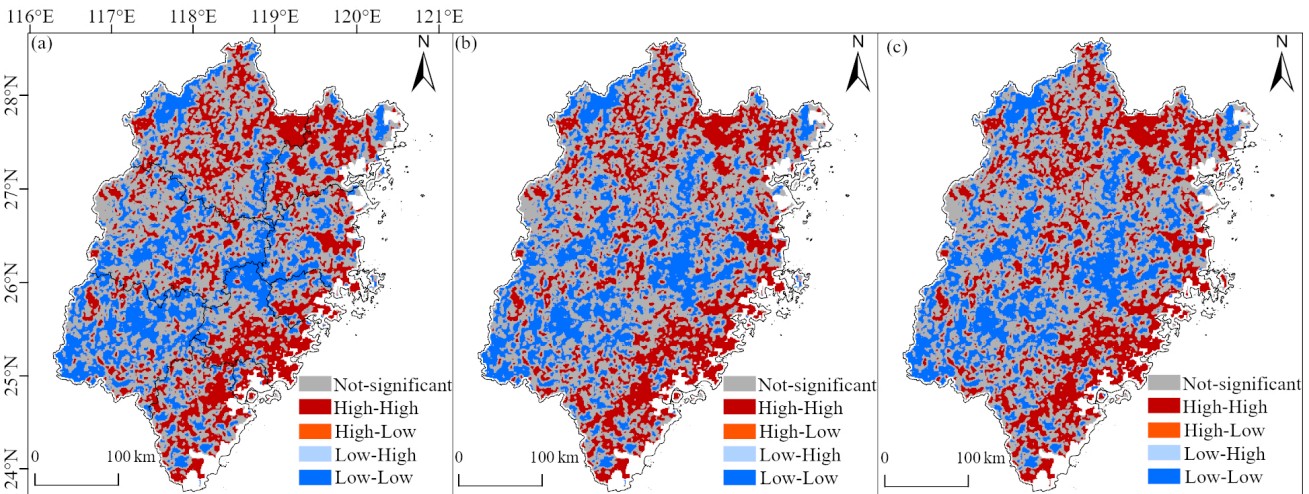

**Figure 6.** Local spatial autocorrelation clustering distribution of FFCI at 1000 m scale: (**a**) 2000; (**b**) 2010; (**c**) 2020.

**Table 6.** Proportions of cluster types at 1000 m scale.

| Type | 2000 (%) | 2010 (%) | 2020 (%) |
|---|---|---|---|
| High–high clusters | 27.2 | 27.0 | 27.0 |
| Low–low clusters | 24.4 | 24.3 | 24.3 |
| High–low clusters | 0.04 | 0.1 | 0.01 |
| Low–high clusters | 0.06 | 0.1 | 0.09 |
| Nonsignificant | 48.3 | 48.5 | 48.6 |

### 3.5. Driving Patterns of Forest Fragmentation Change

#### 3.5.1. Factor Detector

The factor detector was used to determine the explanatory power of various driving factors with regard to the variation in FFCI. We analyzed the q value of driving factors at 11 grid scales (Figure 7), and the q value showed a general increasing trend at a scale of 1000 m to 8000 m for each driving factor, indicating that the explanatory power of driving factors with regard to the variation in forest fragmentation gradually increased in the range of this scale. However, at the scale of 9000 m, the q values of elevation, slope, and NLIV showed a significant decrease, indicating that the explanatory power was weakened. There

was no obvious change in q value of AMTV and APV at scales from 8000 m to 11,000 m. Therefore, the scale of 8000 m was selected as the optimal scale to analyze the impact of driving factors on the variation in FFCI.

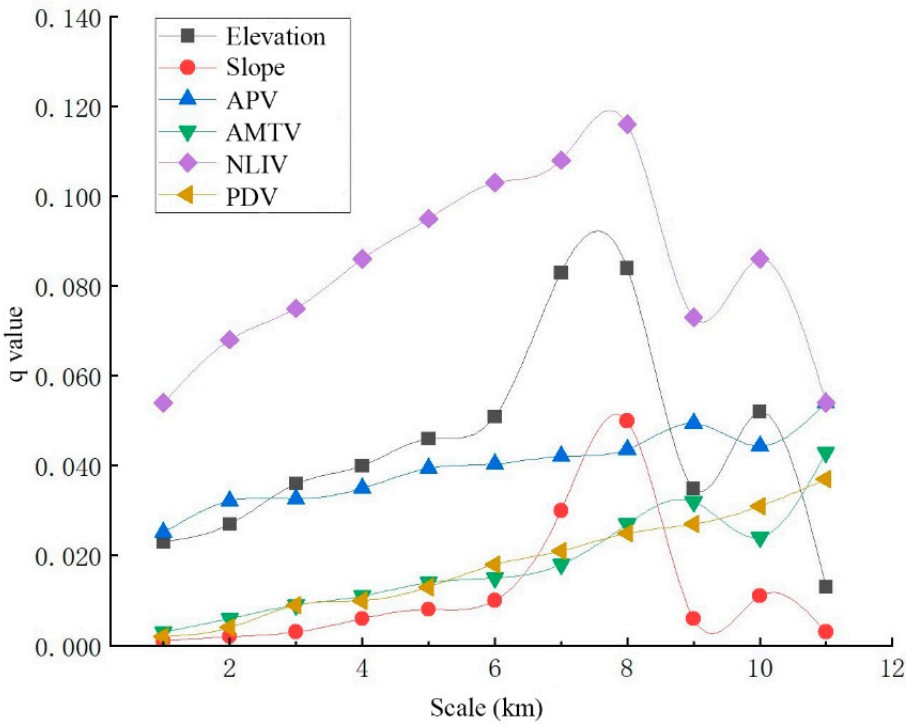

**Figure 7.** Change of scale effect based on geographical detector factor detection analysis.

Figure 8 shows the q value of driving factors at the scale of 8000 m ($p < 0.01$). During periods from 2000 to 2010, the explanatory power of NLIV and elevation was largest among all driving factors, with the q value being 0.116 and 0.084, respectively, while the explanatory power of PDV was lowest, with the q value being 0.025. Results showed that the geomorphic factors and nighttime light intensity were the dominant single factors influencing the variation in forest fragmentation. During the periods from 2000 to 2010, the explanatory power of each driving factor from high to low was elevation > NLIV > slope > PDV > AMTV > APV. Compared to last period, the explanatory power of APV and AMTV decreased, indicating that the impact of meteorological factors on the variation in forest fragmentation was small in Fujian Province.

### 3.5.2. Interactive Detector

With the aid of the interactive detector in GD, we analyzed the superimposed impact of interactions between different driving factors on the variation in FFCI (Figure 9). In the period from 2000 to 2010, the explanatory power with regard to the variation in FFCI by any two driving factors was greater than that of any single driving factors, indicating that the impact of driving factors on spatial heterogeneity of the variation in FFCI was not independent. Specifically, the interaction between APV and slope was the strongest among all interactions (q = 0.174). It can be clearly seen that, while the individual impact of meteorological factors on the variation in FFCI was small, the superimposed impact of the interaction between them and geomorphic factors increased significantly with regard to the variation in forest fragmentation. However, in the period from 2010 to 2020, the interactions between APV and elevation (q = 0.009), and between ATMV and DEM (q = 0.117) were nonlinear weakening, and the explanatory power was weakened; the interactions between slope and APV, and between slope and AMTV were strongest, with q values above

0.157. This indicates that the interaction between slope and meteorological factors jointly contributed to the spatial differentiation of the variation in FFCI from 2010 to 2020.

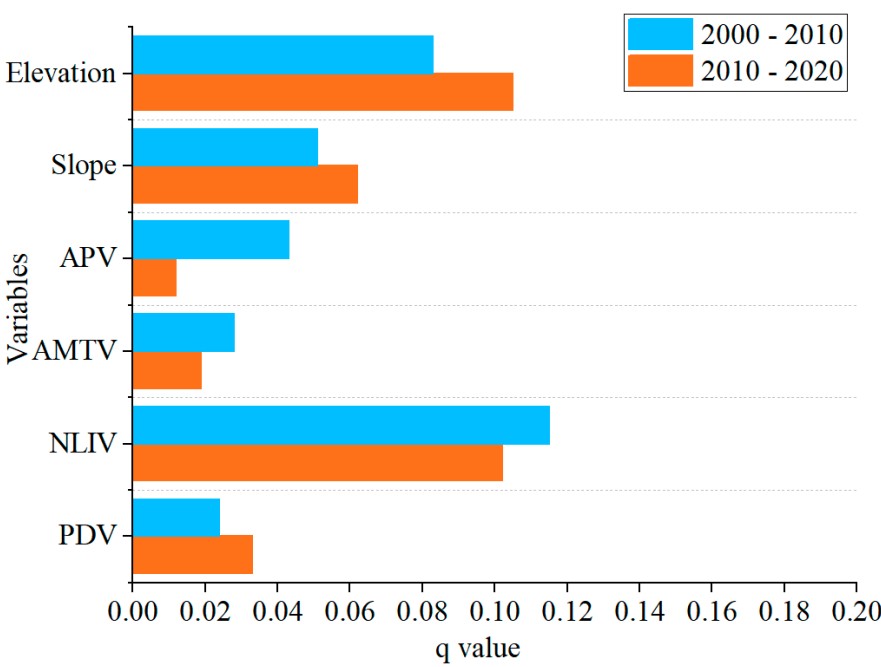

**Figure 8.** The explanatory power of each variable with regard to the variation in FFCI in different periods.

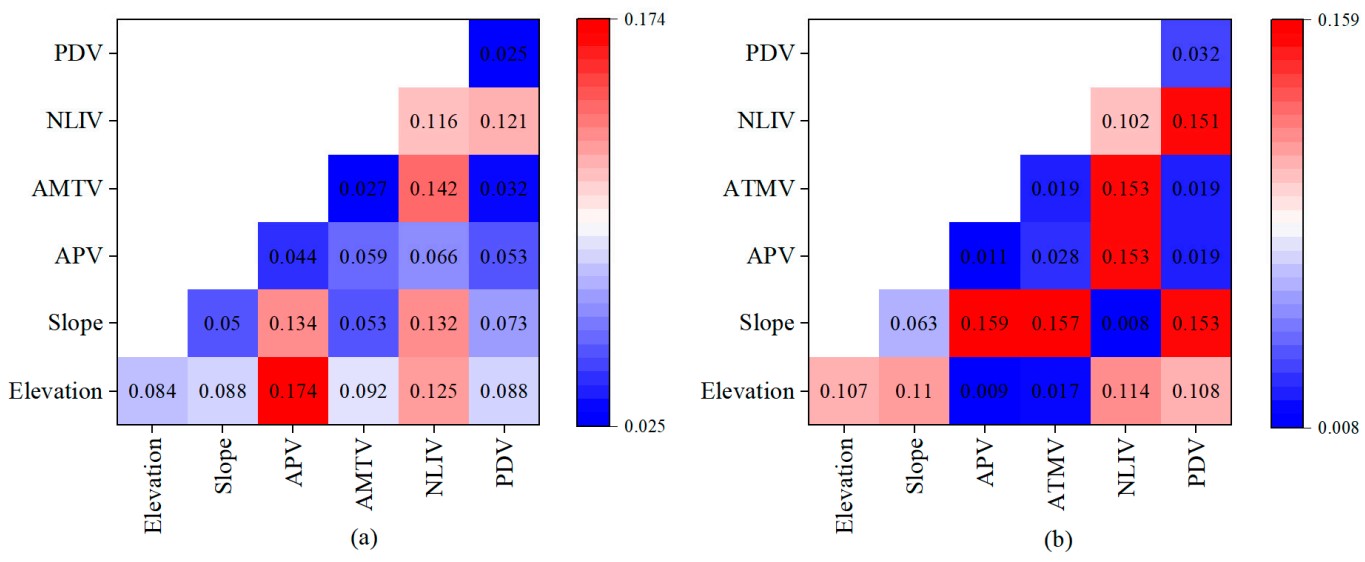

**Figure 9.** The interaction of factors with regard to the variation in FFCI: (**a**) 2000–2010, (**b**) 2010–2020.

## 4. Discussion

### 4.1. Validation of FFCI

An ideal index must be both representative and effective [46,47]. The former requires that the proposed index should comprehensively reflect the characteristics of forest fragmentation. Forest fragmentation is a process in which forest tracts are progressively subdivided into smaller, geometrically more complex patches [19]; this process involves changes in landscape composition and structure. Class-level metrics can quantify the amount, size, and distribution of each patch type in the landscape; thus, they can be considered as indices

for fragmentation [11,17]. We selected five representative metrics (PD, LPI, MPA, AI, and division) to construct the FFCI. To eliminate the multicollinearity among these metrics, PCA was employed to construct a comprehensive index and calculate the weight of each metric in terms of PCs. The PCA method has been commonly used to synthesize comprehensive indices, proving to be adaptive to different regions and various spatial scales [48,49]. In 2000, 2010, and 2020, the percentage variance of PC1 exceeded 85% in all cases with a window size of 3500 m, indicating that the FFCI could explain most of the fragmentation pattern information.

To evaluate the effectiveness of FFCI, validity evaluations were implemented using two approaches: urban–rural gradient-based and transect-based comparisons. For the urban–rural gradient analysis, we computed the mean FFCI along the urban–rural gradients in four prefecture-level cities (Fuzhou, Longyan, Nanping, and Sanming) in Fujian Province. The results showed that the zone closest to the urban center had the highest mean FFCI in all the cases (Figure 10). Urban areas normally have a higher population density and public facility density, and these socioeconomic factors have a great impact on the change of urban landscape [25,50]. With the development and expansion of urban areas, an increasing amount of ecological land (including forests) has been transformed to satisfy the requirement of economic and human activities, resulting in forest patches becoming fragmented. Therefore, frequent land-use change in urban areas normally leads to severe forest fragmentation. Many previous studies also indicated that forest fragmentation is highly correlated with the degree of urbanization and the distance from the urban center [51,52]. These results demonstrate that the FFCI had superior reliability in detecting forest fragmentation.

For the transect analysis, we set up four transects with 15,000 m width (Figure 11a): T1 connected three inland cities, oriented from northeast to southwest; T2 connected three coastal cities, oriented from northeast to southwest; T3 connected inland and coastal cities, oriented from southwest to northeast; T4 connected inland and coastal cities, oriented from northwest to southeast. Sample units were set every 2000 m along the four transects, and the average value of FFCI was calculated (Figure 11b–e). Transect T1 showed that the mean FFCI value in Nanping was higher than that in the other two inland cities, with the mean FFCI value being mostly lower than 0.6 in Sanming and Longyan city. In transect T2, the mean FFCI value usually peaked in the urban center of coastal cities, being significantly higher than the surrounding areas. In transect T3, the curve of the mean FFCI value was flatter, with high values gathered in the urban center of Fuzhou. Transect T4 showed an increasing trend of mean FFCI value from the inland city to coastal city. These results indicated that areas with a low degree of forest fragmentation were mainly distributed in the western and central regions in Fujian Province, while the fragmentation of coastal cities was significantly higher than that of inland cities. As mentioned above, the coastal cities in southeastern Fujian Province have a relatively lower terrain, which means that the human disturbance to the forest landscape is stronger than in internal cities. This may involve high-intensity deforestation and logging, which would exacerbate forest fragmentation. These phenomena are consistent with previous conclusions on the forest fragmentation pattern in China [53], which indicated that forest fragmentation is most severe at low elevations and in developed areas. This also reflects the objectivity of the proposed FFCI index.

### 4.2. The Spatial Patterns of Forest Fragmentation

We used the FFCI to evaluate the forest fragmentation in Fujian Province, observing that the degree of forest fragmentation was above the medium level in Fujian Province during the study period. Although the forest coverage of Fujian province has increased across decades, many studies also indicated that Fujian Province is undergoing severe forest fragmentation [36,53]. This phenomenon can be attributed to forest displacement; whereby deforestation and afforestation occur at the same time, while the locations of forest loss and gain do not coincide. This results in the increased forest cover not necessarily reducing the degree of forest fragmentation. We observed that the areas with high forest

fragmentation were mainly distributed and clustered in coastal areas (Figures 3 and 6), especially in urban centers. This is mainly attributable to the larger scale of urban expansion in China's coastal regions, which has resulted in more severe forest fragmentation [25]. We also found that the degree of forest fragmentation of Nanping was higher than in other inland cities, with high-fragmentation areas mainly distributed in the northeastern and central regions of Nanping. Agricultural expansion and tourism development may have caused this phenomenon in the local area [54], while the forest landscape in western regions has been well protected by the Wuyi Mountain National Nature Reserve. In terms of the temporal variation in forest fragmentation, we found that the degree of forest fragmentation increased from 2000 to 2020, with this trend even more pronounced from 2010 to 2020. This is mainly due to the recent rapid urbanization and infrastructure expansion in China, contributing to the increased forest loss and fragmentation [55].

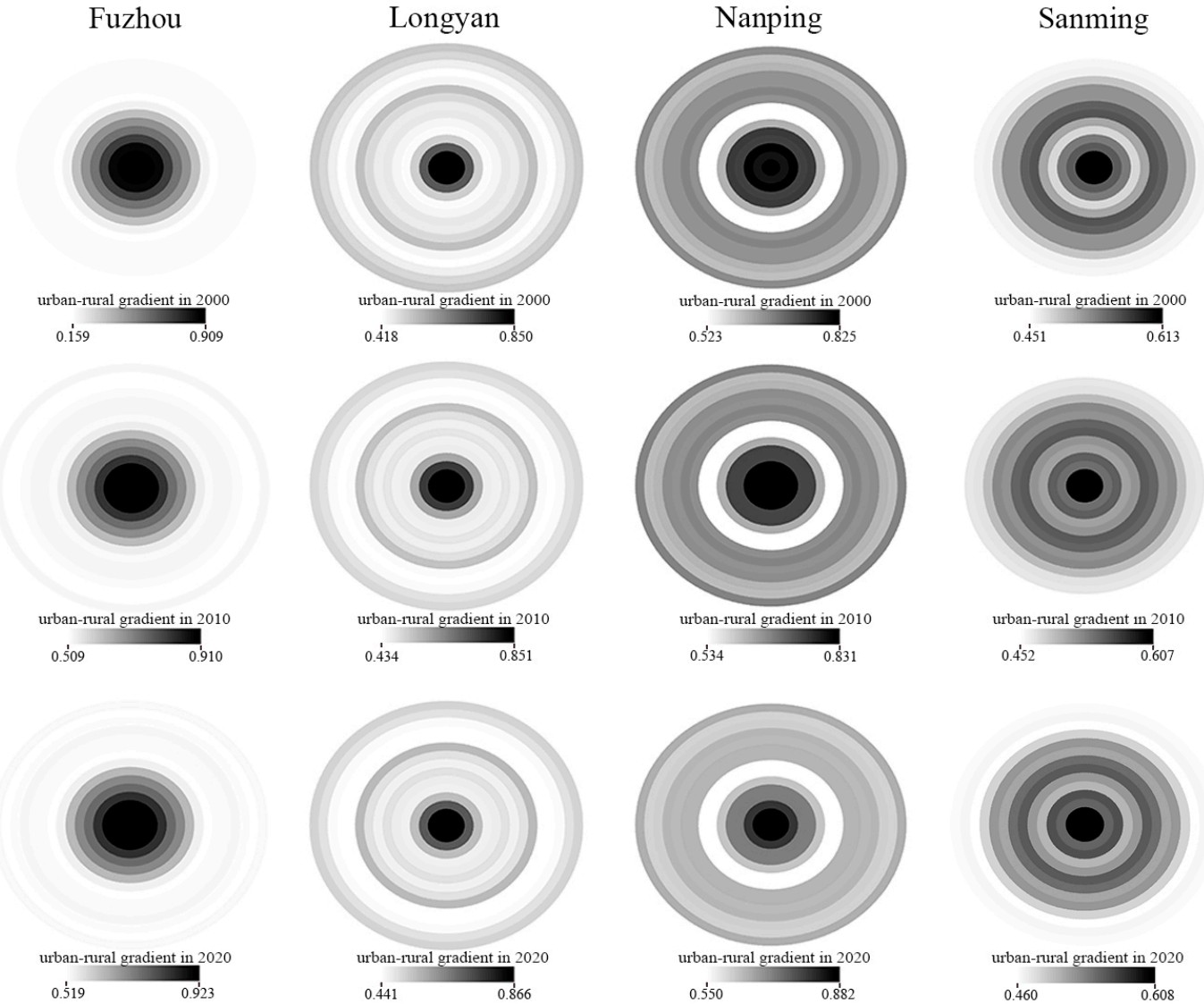

**Figure 10.** Mean FFCI along the urban–rural gradients for representative cities. The center of the circle is the location of the city government, while the radius represents the distance from the urban center (2000 m, 4000 m, 6000 m, . . . , 30,000 m).

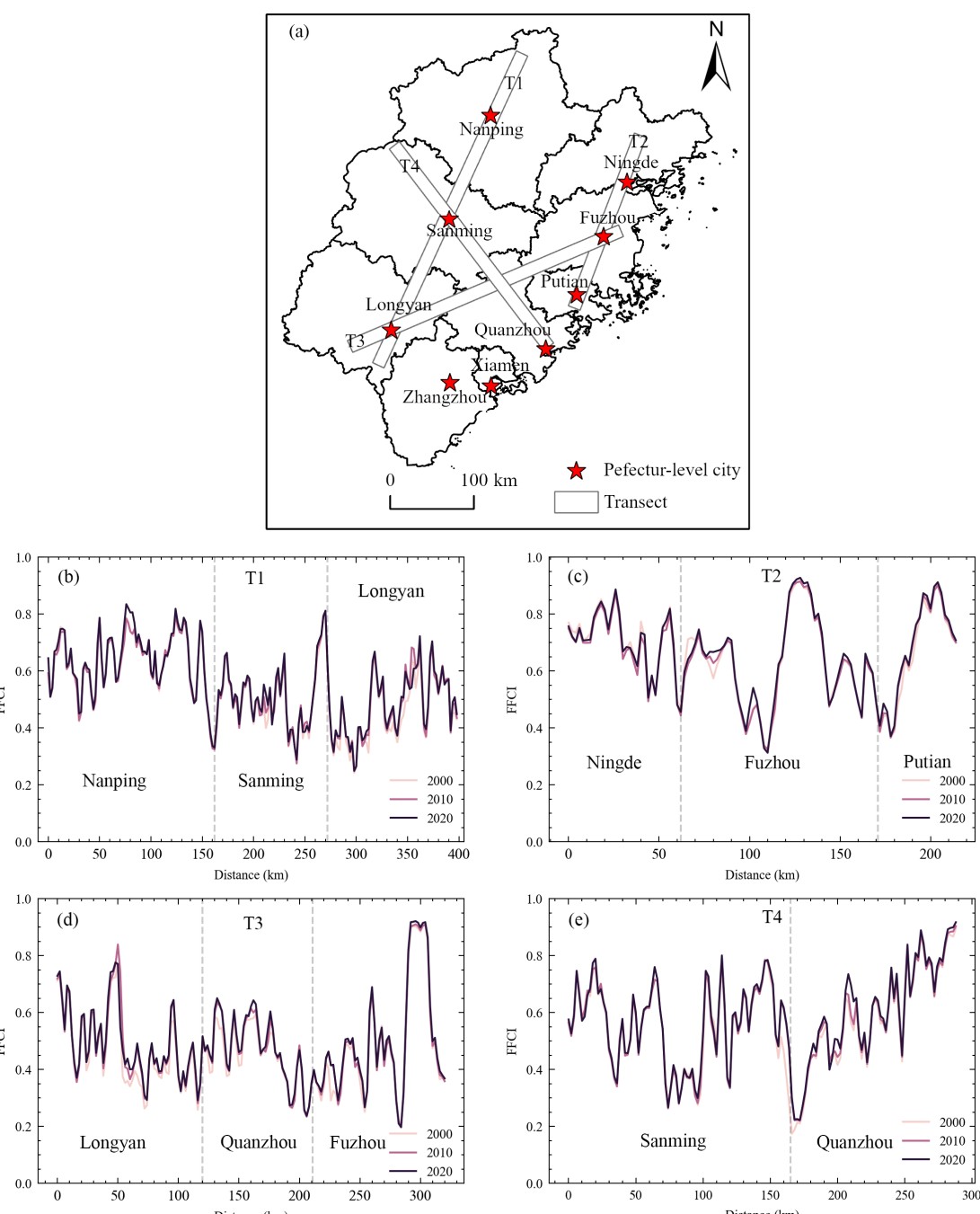

**Figure 11.** The location of transects with 15,000 m width along with the corresponding FFCI changes: (**a**) location of transects; (**b**) FFCI changes of transect T1; (**c**) FFCI changes of transect T2; (**d**) FFCI changes of transect T3; (**e**) FFCI changes of transect T4.

*4.3. The Driving Pattern of Forest Fragmentation Dynamics*

Forest fragmentation is normally influenced by socioeconomic, nature, and anthropogenic factors. In this study, geomorphic factors (i.e., elevation and slope) were the most prominent driving factors leading to the variation in forest fragmentation, similar to the results in a previous study [56]. This is because gentle and low-elevation areas are suitable for urban development and human activity, making forests vulnerable to occupation and transformation into other types of land, which increases the degree of forest fragmentation. However, due to the decrease in human disturbances, there are no significant changes to forest fragmentation in steep and high-elevation areas.

As an indicator of human activity, the variation in nighttime light intensity is also a major driving factor of variation in forest fragmentation. The variation in nighttime light intensity can intuitively reflect the urbanization process [57]. Previous studies have indicated that rapid urbanization is associated with a change in urban green areas, resulting in the high fragmentation of urban green spaces [58].

The single impact of the variation in temperature or precipitation on the variation in forest fragmentation was small, which is similar to the conclusion by Chen et al. [59]. This is mainly due to there being sufficient natural conditions for vegetation growth available in Fujian. However, we also found that the interaction between meteorological factors and other factors could significantly amplify the impact on forest fragmentation in Fujian Province, indicating that multiple factors jointly drive the variation in forest fragmentation.

## 5. Conclusions

In this study, we constructed a comprehensive index (i.e., FFCI) to evaluate the extent of forest fragmentation, and we analyzed the spatiotemporal pattern of variation in forest fragmentation in Fujian Province, as well as explored the driving factors of changes in FFCI using GD. The conclusions are presented below.

The FFCI was synthesized using the PCA technique on the basis of five landscape metrics: AI, LPI, MPA, division, and PD. PC1 with a window size of 3500 m could explain more than 85% of the information from all selected landscape metrics in the study period. Using the proposed FFCI, we found that the degree of forest fragmentation in Fujian was at an upper–middle level. In 2000, 2010, and 2020, the proportion of areas with high fragmentation ($0.6 \leq$ FFCI $< 0.8$) and medium fragmentation ($0.4 \leq$ FFCI $< 0.6$) was highest, and the sum of these two proportions exceeded 50% in each year.

In terms of the temporal change of forest fragmentation, from 2000 to 2010, there was a decrease in forest fragmentation in 2.7% of areas, whereas the areas with increased forest fragmentation accounted for 4.8%. Quanzhou, Zhangzhou, and Longyan were the primary areas where forest fragmentation increased, while Ningde had the largest decrease. From 2010 to 2020, the areas with increased fragmentation accounted for 6.7%, a slight increase from the previous stage. The proportion of the areas with reduced forest fragmentation was 2.8%, a slight decrease compared with the previous stage.

The spatial difference in FFCI was mainly characterized by high–high and low–low clusters. The high–high clusters were mainly distributed in coastal areas, Nanping and Ningde, mostly concentrating in the central urban areas of coastal cities. On the other hand, the low–low clusters were mainly distributed in the western and central regions of Fujian Province, including Longyan, Wuyishan, and the junction of Quanzhou and Fuzhou.

The explanatory power of each driving factor with regard to the variation in FFCI from high to low in 2000–2010 was ranked as NLIV > elevation > slope > APV > AMTV > PDV, and the interaction between APV and elevation had the greatest enhancement effect among all interactions (q = 0.174). From 2010 to 2020, the explanatory power of each driving factor was ranked as follows: elevation > NLIV > slope > PDV > AMTV > APV; the interaction between slope and APV had the most obvious enhancement effect (q = 0.159).

**Author Contributions:** Conceptualization, X.H.; methodology, X.H. and J.L.; software, Q.Z. and S.Z.; formal analysis, Z.W. and S.Z.; data curation, Q.Z. and S.Z.; writing—original draft preparation, S.Z.; writing—review and editing, S.L. (Sen Lin) and X.H.; visualization, S.L. (Shuang Liu) and S.L. (Sen Lin); supervision, J.L. and X.H. All authors have read and agreed to the published version of the manuscript.

**Funding:** This research was funded by the National Natural Science Foundation of China (No. 31971639), the Natural Science Foundation of Fujian Province (No. 2019J01406), and the Forestry Peak Discipline Construction Project of Fujian Agriculture and Forestry University (72202200205), to which we are very grateful.

**Institutional Review Board Statement:** Not applicable.

**Informed Consent Statement:** Not applicable.

**Data Availability Statement:** The data are contained within the article, and all data sources are mentioned.

**Conflicts of Interest:** The authors declare no conflict of interest.

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
