# Peer review of "Detecting Spatiotemporal Dynamics and Driving Patterns in Forest Fragmentation with a Forest Fragmentation Comprehensive Index (FFCI): Taking an Area with Active Forest Cover Change as a Case Study"

_forests, doi:10.3390/f14061135_

Round 1
Reviewer 1 Report
This paper describes the construction of a forest fragmentation comprehensive index through PCA for 5 land use data to identify patio-temporal variations characteristics of forest fragmentation in Fujian, to determine the dominated driving factors on variations of forest fragmentation, and to identify the impact of the interaction between driving factors. The content is clearly described, and the study is highly replicable.
This study uses the first principal component as the overall forest fragmentation index because of the high contribution of the results of principal component analysis. While there is nothing wrong with using an index with a high contribution rate in a principal component analysis, there is a major problem with comparing the first principal component obtained from data sets from different time periods. This is because the weights (coefficients) for each variable in the first principal component obtained in a principal component analysis for data sets from different time periods are different for each variable and are likely to take different forms of disaggregation even when the first principal component values are the same. Thus, in an analysis based on multiple measures of fragmentation, it is not possible to know exactly what the comparisons using the values as indicators represent, since the definition of fragmentation differs from one data set to another.
While "2.3 Calculation of forest fragmentation comprehensive index" describes which landscape metrics were selected for principal component analysis, it does not describe how to construct the FFCI." In "3.2 Spatio-temporal characteristics of FFCI," the results of the principal components analysis show that the contribution rate of the first principal component exceeded 85% in all cases, so this is taken as the FFCI. However, it does not state how much or more the contribution rate of the first principal component should be to be considered as FFCI. It also does not describe how to construct the FFCI if it is below the threshold. Therefore, even though the purpose of this paper is to construct the FFCI, it is not clear whether the FFCI can be determined when the data set changes, for example, when there is a large-scale development in another area or even in the same area.
L114 MSS should not have been used in this data. Also, ETM is correct for ETM+.
L131 Monthly precipitation should be preceded by a line break.
L157, L179 It is difficult to understand the difference between optimal analysis scale and optimal spatial scale.
L193-L194 It is necessary to explain what "high" and "low" mean and by what criteria they are distinguished.
L240-242 It is necessary to explain how this can be said.
L270-L271 Does it represent a change in Fragmentation type? It is strange that with FFCI itself almost everything is Unchanged.
Reviewer 2 Report
The manuscript is very interesting. The methods and analyses were well applied bringing excellent results. I think it is important to give more emphasis (including in the abstract) to the interesting fragmentation patterns described, which are expressed in the discussion. In this sense, it would be interesting to make the abstract more palatable to a wider audience and to highlight a little more the "biological" or "environmental" meaning of the results.

Reviewer 3 Report
The study created a comprehensive index to assess the extent of forest fragmentation and investigated the driving factors of change dynamics as well as analyzing the sptio-temporal variation pattern of forest fragmentation in Fujian province.
The article's title is suitable with the content of the paper. The abstract is well-designed and briefly express the present research thus being of interests and readable thus capturing the reader's attention. It present in an appropriate manner the main research hypothesis, the problem statement, the key words are appropriate to the present research and are clearly stated.The structure of the paper is correct in line with the journal standards.
The study methodology needs to be explained in a little more detail.
The main research is well design and appropriate conducted in line with the main questions in spatial planning in the investigated areas.
I think the English is ok as far as I could see but I am not a native speaker.
Also, I have a few suggestions to the author:
Landsat image was used in the study. Do you think the resolution of these images is sufficient? It seems to me that the 30 m resolution may not allow you to do very precise work? Which data types were used in the studies in the literature, Landsat, Sentinel, maybe airbone LIDAR? I suggest you add an explanation about it.
Introduction and Methodology section should be developed a little more. Results
Or I recommend the interpretation of the reasons for the results in the discussion section. so it will be more informative for readers, and will attract more attention. For example: it should be explained that forest fragmentation is highly correlated with the degree of urbanization and distance from the city centre. This is just an example. The findings require further comment. The figures are very understandable. In the conclusion part, the general results of this study should be given more place.
Reviewer 4 Report
Dear authors,
The manuscript entitled: “Detecting spatio-temporal dynamics and driving patterns in forest fragmentation with a forest fragmentation comprehensive index (FFCI): Taking an area with active forest cover change as a case” was evaluated. The assessment was carried out very carefully.
The observations will be simple, however it has importance in its configuration.
1. Standardize the units of measurement presented throughout the text of the manuscript;
2. The location map could be improved, in which there is no caption for some highlights;
3. In the satellite methodology, observe other missing features, such as temporal resolution;
4. Also in the methodology, draw the equations in the text of the manuscript;
5. I also believe it is necessary to use the geographic coordinates in the thematic maps presented in the results;
Other observations are in the attached manuscript.

Round 2
Reviewer 1 Report
The authors follow the comments and judge that they have been carefully revised.